# Adolescents’ Knowledge and Misconceptions About Sexually Transmitted Infections: A Cross-Sectional Study in Middle School Students in Portugal

**DOI:** 10.3390/healthcare12222283

**Published:** 2024-11-15

**Authors:** Catarina Abreu, Luísa Sá, Paulo Santos

**Affiliations:** Department of Community Medicine, Information and Health Decision Sciences (MEDCIDS), Center for Health Technology and Services Research (CINTESIS@RISE), Faculty of Medicine, University of Porto, 4200-393 Porto, Portugal; up201807377@edu.med.up.pt (C.A.); luisadesa@med.up.pt (L.S.)

**Keywords:** adolescent, sexuality, sexually transmitted diseases, sex education, health education

## Abstract

**Background/Objectives**: Adolescence represents a period of heightened vulnerability to risky sexual behaviours. In Portugal, adolescents are initiating sexual activity at increasingly younger ages, making it relevant to characterise their knowledge about sexually transmitted diseases and prevention methods at an earlier stage of their development. **Methods**: A cross-sectional observational study was conducted involving the application of a questionnaire to 750 ninth-grade students from Vila Nova de Gaia, Porto, Portugal. The questionnaire covered topics such as perceptions of sexuality, knowledge of sexuality and STIs, methods of transmission prevention, and sources of information. **Results**: The mean age was 14.4 years, with 19.9% reporting having already initiated sexual activity. Overall, the knowledge about sexually transmitted infections was low, with a score of 7.08 out of a maximum score of 18. Condoms and immunisation were well-recognised preventive measures, but many misconceptions persisted. Better knowledge is associated with better attitudes regarding the likelihood of having sexual intercourse. **Conclusions**: This study reveals adolescents’ limited knowledge about sexually transmitted infections, misconceptions about prevention, and reliance on peers and the internet over trusted sources, highlighting the need for comprehensive sexual education in middle school.

## 1. Introduction

Adolescence is a crucial stage in individual development, as it is a period of learning and clarifying values. During adolescence, we build our personality and begin to distinguish ourselves as individuals. Most of this development arises from self-assertion and experimentation, which exposes young people to vulnerability and risky behaviours, particularly in the realm of sexuality [1]. Unprotected sex and other risky sexual behaviours pose a severe threat to the physical and psychological health of adolescents, as well as their social well-being [2].

Despite the vast heterogeneity found in studies, in recent years, the onset of sexual intercourse seems to occur at increasingly younger ages [3], without being accompanied by consistent sexual education [4]. The concept of “early onset of sexual activity” is commonly defined as a first sexual encounter before the age of 15 [5], usually associated with poorer outcomes in terms of sexual health [6]. In Portugal, several studies have been conducted on adolescent sexuality in recent years, indicating mean ages for the first sexual encounter from 15.5 to 16.7 years old [1,7,8], concordant with the international literature, posing adolescents at an increased risk of unwanted pregnancy and sexually transmitted infections.

The World Health Organization (WHO) estimates that there are over 1 million new diagnoses of sexually transmitted infections (STIs) daily. Adolescents have the highest incidence rate and have shown stable total STI trends since the 1990s [9]. This significant proportion of young people in the incidence of STIs highlights a gap in sexual education for this age group, which has severe repercussions on their health and well-being. It is concerning that knowledge often stems more from the greater visibility that some STIs assume in the media than from targeted sexual education [1], promoting a more casuistic response model rather than adequate literacy, especially among younger individuals with less emotional maturity. Lower knowledge is associated with worse attitudes and skills, measured by lower utilisation of condoms [7,10], primarily due to method unavailability and false beliefs, and early onset of sexual intercourse.

Therefore, literacy emerges as a fundamental factor in influencing decision-making capacity in health, particularly in sexuality. Informed youth are more aware and capable of making free and appropriate health decisions based on the best evidence [11]. Sexual education at young ages promotes critical thinking, problem-solving empowerment, and confidence for informed choices [7].

Health literacy is influenced by available sources of information, notably the internet and social media. The internet allows for easy and rapid access to information worldwide, in real time and privately. It is an increasingly utilised tool among young people, who seek and share health-related information, exchange social support in online communities, and track and share health statuses or activities [12]. However, it presents concerns about the quality of information and the lack of scientific verification that limit its usefulness and reliability [13,14], where interpersonal information wins, particularly from family, friends, or healthcare professionals, leading to the need to change the health services offer to get closer the younger, using the communication tools they also use inclusively, particularly in family planning issues [7].

Prevention is to anticipate. Anticipating in this area is to search for the expectations, ideas, and knowledge prior to the onset of sexual activity. In Portugal, we have a strong strategy for HIV prevention based on condom utilisation and testing. Vaccines against HPV and HVB are included in the national immunisation program, giving them importance for individual protection and social relevance. Recently, Portuguese authorities published recommendations about screening for gonorrhoea and chlamydia infections. As far as we know, there are no studies in Portugal addressing knowledge about STIs in general and prevention methods among adolescents in middle school. Adolescents under 15 years old tend to be more prone to less safe choices in their relationships and sexual practices, making it crucial to characterise their knowledge before it begins [6]. Are their attitudes merely influenced by the impulsivity typical of their age, or are they genuinely uninformed about the risks they face? What are their sources of information? What are the myths and misconceptions related to sexual behaviours in this population?

We aim to characterise the adolescents’ knowledge and misconceptions about STIs, including their determinants and prevention strategies. We want to characterise their vision about sexuality, sources of information, and misconceptions to better define standards for the improvement of education interventions on this topic.

## 2. Materials and Methods

We conducted a cross-sectional study by presenting a written self-answered questionnaire to 9th-grade students from the municipality of Vila Nova de Gaia in the Northern Region of Portugal. This municipality is one of the most densely populated in the country, with approximately 303,824 inhabitants, of which 12.8% are young people under the age of 15.

### 2.1. Sampling and Participants

All public schools in the municipality were invited to participate in the study, corresponding to an eligible population of 1313 adolescents attending the 9th grade. Among the eighteen schools invited, seven agreed to participate in the study. No exclusion criteria were established.

The survey was conducted between September and October 2018.

### 2.2. Questionnaire

The questionnaire used in this study was previously tested in a Portuguese survey [15], in which the respondents were adolescents and young adults.

The questionnaire consists of 4 parts: questions regarding sexuality (values and level of knowledge), questions regarding knowledge about STIs, questions about sources of information, and general demographic characterisation data.

We used the following variables in the analysis: demographic characterisation data (age and sex), data on sexual behaviour (sexual activity, number of sexual partners in the past year, and previous history of STIs), self-perceived knowledge about sexuality, the definition of sexuality, self-perceived knowledge about STIs and strategies for prevention, attitudes towards prevention considering the chance of a sexual encounter, and sources of information on sexuality and STIs, including frequency of use and importance.

To define sexuality, participants were given 14 options of words/expressions from which they had to select the three options they identified with the most, as described by Abreu (2010) [16]. Likert scales were used to assess self-perceived knowledge about sexuality, STIs, and strategies and attitudes for prevention of transmission. Numeric scales from 1 to 3 were used to determine the frequency and importance of the sources of information that young people turn to for advice on sexuality and STIs.

### 2.3. Data Collection Procedures

The questionnaire was distributed in paper format in the classroom for all the students who agreed to participate and had all the necessary authorisations. Even with the tutors’ permission, they could choose not to participate without having to justify it and without this decision affecting their regular academic activities.

An alphanumeric code identified the questionnaires. The alphabetical component identified the educational institution, allowing for the determination of the response rate in each school. The numerical component was consecutive and served only for quality control of record transcription. Responses were completely anonymous, and retrograde identification of participants was not possible.

### 2.4. Ethical Considerations

The present research study adhered to applicable laws and regulations of the Helsinki Declaration principles and the Oviedo Convention. The Ethics Committee of the Faculty of Medicine, University of Porto, approved the protocol and further procedures (process number 225/CEFMUP/2023).

The administration of the schools and parent associations approved the study, and individual informed consent was obtained from parents and legal guardians.

### 2.5. Statistical Analysis

The data were recorded in Microsoft Office 365 Excel^®^ format, encoded, and analysed by using IBM SPSS Statistics 26^®^. We used descriptive statistics as means and medians for continuous variables and proportions for categories. The confidence intervals used 95% certainty by Wald’s formula.

The knowledge score resulted from the respondents’ current knowledge about STI prevention methods, which was as high as the number of correct answers selected. Methods deemed effective for STI transmission prevention included condom use, abstinence from sexual intercourse, male circumcision, and vaccination against Hepatitis B and HPV. Participants who agreed or strongly agreed scored one point. Other methods were considered ineffective and scored one point if the participants selected strongly disagreed or disagreed. All other options, including “Don’t know/Don’t want to answer” and missing, were considered a knowledge deficit, resulting in zero points on the score. Results were analysed by the proportion of correct answers for each question. Due to the normal distribution of results, the score was dichotomised based on the median obtained: “higher knowledge” (score > 7) and “lower knowledge” (score ≤ 7). This dichotomisation was used for logistic regression analysis purposes.

All the variables were treated from a positive point of view. Participants presented knowledge if they answered affirmatively that they knew. Missing and “do not know” or “do not want to answer” were treated as a deficit in knowledge and assumed to be negative.

The sources of information were characterised by relevance and frequency of utilisation, using a 3-point Likert scale, in which the mean was transformed into a percentage for analysis. The association between frequency of utilisation and relevance was determined by the correlation of Kendall τ.

Similarly, the attitudes towards the likelihood of a sexual encounter were analysed as the proportion of adolescents who declared them to be important or very important in the total sample after categorising the 1 to 5 Likert scale and assuming the missing and no responses as negative attitudes.

A multivariate logistic regression model was used to describe the factors associated with better knowledge about sexually transmitted diseases, using age and sex as confounding variables. Also, a multivariate logistic regression model adjusted for age and sex allowed us to quantify the relationship between knowledge, dichotomised in high or low according to median distribution, and the attitudes regarding the likelihood of a sexual encounter.

We accepted an alpha error of 0.05.

## 3. Results

Seven schools in the municipality of Vila Nova de Gaia agreed to participate in the study. A total of 750 students from the ninth school year answered the questionnaire (response rate of 78.3%). The remaining 208 students were not included because they needed to present the authorisation of their guardians, missed classes on those days, or expressed the intention not to participate before or while filling out the questionnaire.

The mean age was 14.4 years old (±1.0), and 51.4% were females (Table 1). A total of 121 (18.9%) reported having already started sexual activity, mostly males (n = 67), with an average age of 15.2 ± 1.2 years. In the sexually active group, 58.3% declared to have had one partner in the last year, and 35.9% had had more than one. Only three participants (0.4%) reported a history of a sexually transmitted disease in the past, two females and one male.

About two-thirds considered to have good or very good knowledge about sexuality issues (64.6%), with males higher than females, but less than half considered to have good or very good knowledge about sexually transmitted infections (45.9%), without differences between sexes.

The most used words to describe sexuality were love, respect, and pleasure. Females tended to identify with respect, health protection, and self-esteem more than males, who identified more with pleasure and beauty (Table 1).

Regarding the perceived level of knowledge about different STIs (Table 1), the diseases that respondents reported knowing better were HIV/AIDS infection (mean of 4.09 ± 1.01), genital herpes (mean of 3.04 ± 1.34), hepatitis B (mean of 2.78 ± 1.31), and hepatitis C (mean of 2.73 ± 1.33). Trichomoniasis (mean of 1.05 ± 1.30), chlamydia infection (1.09 ± 1.41), gonorrhoea (mean of 1.83 ± 1.46), and syphilis (1.69 ± 1.54) were the diseases with the least reported knowledge. Females reported higher knowledge about hepatitis B and C, human papillomavirus, and vaginal candidiasis, and males reported better knowledge about syphilis and gonorrhoea. We found no differences between sexes in human immunodeficiency virus (HIV) infection, genital herpes, chlamydia infection, and Trichomonas infection. Table 1 shows the distribution of answers for each disease.

The use of condoms and vaccination against HBV and HPV were the preventive measures for STIs most correctly recognised by adolescents, followed by practising only anal, oral, or vaginal sex. There was a rate of correct responses of less than 50% in the remaining preventive measures, with “having recent normal analyses” and “regularly performing Pap smears” being most wrongly considered as effective preventive activities (Figure 1). Globally, the total knowledge score was 7.08 (95% CI: 6.82–7.33), out of a maximum score of 18, with 56.0% being below the mean value and only 4.5% ranking above 80% of the total. Although significative, the correlation with self-perception of knowledge was very low (Kendall τ = 0.117; *p* < 0.001).

Regarding the relevance given to several attributes of the partner, thinking about the likelihood of having sexual intercourse, adolescents valued more the level of knowledge they had about their partner, the actual availability of condoms and regular contraception, and the absence of diseases both reported by their partners and verified by the presence of wounds or redness. There were several differences between genders (Figure 2). Females valued knowing the civil status of their partner (66.8% vs. 51.5%, *p* < 0.001) and taking regular contraception (76.2 vs. 66.8%, *p* = 0.004), while males valued knowing about previous partners (49.6 vs. 38.0%, *p* = 0.001) and the general appearance (51.5 vs. 36.9%, *p* < 0.001).

We studied the sources of information by the frequency of utilisation and the relevance of each source (Figure 3). Adolescents attached great importance to their parents and healthcare, although the frequency of utilisation was relatively low. The higher correlation between frequency and relevance occurred for the internet (Kendall τ = 0.630) and social networks (Kendall τ = 0.626). The lowest was found for healthcare providers (Kendall τ = 0.370) and parents (Kendall τ = 0.401), meaning low utilisation despite their importance.

We found several factors associated with better knowledge: being sexually active; higher self-perceived knowledge about sexuality issues and STIs; describing sexuality as love; greater knowledge about HIV infection (AIDS), genital herpes, hepatitis B virus, human papillomavirus, syphilis, and gonorrhoea; and turning to friends as a frequent source of information. On the other hand, defining sexuality based on self-esteem (having positive self-feelings, such as trust and appreciation) was related to lower knowledge (Table 2).

We prospected for the association between knowledge and attitudes toward the likelihood of having a sexual encounter. Better knowledge was associated with higher importance given to actual condom availability, taking regular contraception, asking for a recent negative HIV test, checking for the presence of lesions in the genital area, and lower relevance of the partner’s general appearance or belonging to a church (Table 3).

## 4. Discussion

In this 14-year-old population at the end of middle school (ninth grade), the utilisation of condoms and immunisation against HPV and HBV were the principal measures identified as preventive for STI transmission, following the relevance attributed by a national strategy for the fight against STIs. The self-perception of knowledge about HIV infection is linked to better knowledge, with a higher association. Our outcomes reflect this entire continuous process of education and changes in attitudes towards health and are consistent with previous studies [17,18].

However, adolescents failed to identify all other factors. For only 5 of the 18 questions, there were more than 50% of participants able to answer correctly. The global evaluation of the knowledge about STIs in this population was very low, with a mean of 7 and a maximum of 18 points, meaning a significant progression margin to improve their knowledge and attitudes regarding sexuality and STI prevention. Myths and false beliefs, such as the ability of bathing or genital washing after sexual intercourse and the use of antiseptics and oral contraception to prevent STIs, are still very common in this population. Education programs must address specifically these misconceptions to really change attitudes and skills in sexual behaviours, which is crucial to improving the current situation [18,19].

The STIs with the highest self-perceived knowledge were HIV/AIDS, genital herpes, and hepatitis B, which is consistent with previous studies [1,18,20], explained mainly by the major exposure to public opinion, the internet, and the media, having as a result an increasing awareness regarding these infections in the last decades [21], although it is possible that there is some confusion between type 1 and type 2 herpes virus [1]. As in other studies [1,18,20], trichomoniasis, chlamydia infection, and gonorrhoea were the least known diseases, despite their notification rates presenting increasing tendency in recent years [22]. This proves the need for better health education covering real-world epidemiological patterns and focusing on skill improvement, more than allowing one to base all contents on some paradigmatic disease, even though it is of specific importance, such as the case of HIV.

As expected, being sexually active is associated with better knowledge, probably due to the need to solve practical problems when they actually arise, functioning as a promotor of problem-based learning [1]. Sexual activity is then a driver to seek information based on one’s own experiences and an opportunity for discussion. The need for good sources of information is crucial, since there is a low correlation between what they know and what they think they know.

We found a significant association between identifying “love” to describe sexuality and better STI knowledge. On the other hand, “self-esteem” was associated with lower knowledge. The way adolescents look at their sexuality appears to influence their seeking information and their knowledge about STI prevention. Adolescents who approach sexual relationships with a focus on emotional connection, love, and intimacy tend to have greater closeness and more profound affection with their sexual partners, which could lead to better openness to communicate and discuss their doubts and concerns regarding sexuality and STIs. On the contrary, adolescents who identify their sexuality as “self-esteem” may view their sexual experiences through the lens of self-worth, confidence, and personal validation, which could lead to a lesser focus on external factors such as partner communication, safer sexual practices, and STI prevention [16]. These adolescents may perceive seeking information about sexual health as a threat to their self-image or feel uncomfortable discussing topics related to STIs due to concerns about judgment or stigma.

Although two-thirds referred to having good or very good knowledge about sexuality issues, primarily males, their self-evaluation about their knowledge of STIs was lower, in line with our previous study [15], without differences between genders. At this age, the conceptual framework of sexuality does not translate into practical issues regarding disease prevention. In an older population of Portuguese adolescents, Silva found better knowledge, with about 80% of adolescents considering their knowledge of STIs and contraceptive methods good or excellent [1].

We found that the most relevant sources of information about STIs were healthcare providers, parents, and partners. However, despite being included in the most frequent sources, their utilisation was relatively lower, with friends and the internet appearing at the same level. Friends present the greatest impact on knowledge, meaning they are a powerful force for behaviour change if the peer group is well directed [23].

Adolescents seem to believe that parents, healthcare providers, and even their teachers are trustworthy in their knowledge but distant in the face of perceived needs and, therefore, less used. The internet is an interesting point that deserves discussion. It has straightforward accessibility and does not require any specific structuring, both from a formal point of view in the queries used for browsing and from a symbolic point of view in terms of the relational and moral constraints potentially associated with some questions, especially in sexuality. However, there are multiple biases in the information conveyed on networks, which may compromise trust. In general, in Portugal, there is great concern about the veracity of the information published on the internet. Still, at the same time, the levels of trust in the information are also high in a particular cultural pattern [24]. This is an aspect to be considered, and it leads us to the need to have credible agents managing the available information so that it is valuable and reliable. Our results tell us that these agents, in this particular age, are mainly parents and health professionals who, more than linking information, must prescribe the available tools, distinguish quality, and influence adolescents’ choices [25]. Friends and group behaviours play a relevant role in shaping one another’s beliefs, attitudes, and behaviours, including those related to sexual health. Communicating and discussing sex practices and experiences could be beneficial for these adolescents, as it generates curiosity and knowledge-sharing opportunities in a comfortable and relatable setting. Peer-to-peer conversations are more accessible and less intimidating than those with teachers or healthcare providers, and they can play an essential role in promoting STI prevention knowledge. The school environment may present good opportunities to enhance this potential, promoting peer discussion, backed by teachers or local health providers, based on sound scientific evidence and ensuring a quality discussion capable of creating knowledge and skills. On the other hand, religiosity seems to play just a marginal role in sexuality knowledge and STI prevention. The literature points out that religious involvement is associated with more conservative sexual attitudes, with an impact on later sexual debut; more contraceptive utilisation [26]; and less self-assessed risk of STIs [27]. In adolescents, however, this effect seems to be less pronounced [28], perhaps because of their tendency to demise from the Portuguese socio-cultural identity of the Catholic matrix [29], based on the remoteness of their expectations from the moral positions of celibacy and monogamy that move away from the experimentation typical of adolescence towards a more regulatory and punitive stance contained in the message of the sin. A more integrative message may produce better results and take advantage of the forums of young people who still come together today around religious experiences.

Higher knowledge is associated with better attitudes toward daily choices, although there is still too much weight on the confidence of the partner and what they know about them. In this population, almost one-fifth of adolescents reported having already initiated sexual activity, especially among males. However, we cannot be sure that this sexual activity corresponds to explicit sexual intercourse or just intimate contact. Silva et al. found a proportion of 22.2% of adolescents with sexual activity onset before the age of sixteen [1]. As in other studies, males also present a tendency to have more sexual partners [7,30].

We studied attitudes by the importance given to several factors to consider in the hypothesis of a sexual encounter. Adolescents reported valuing the level of knowledge they have about their partner, the current availability of condoms, and whether the partner reports not to have an STI. Ferreira found an impressive 98% of young people (aged 15–19 years) who recognised the risks of unprotected sex and the role of condoms, although about one-fifth did not systematically use them [7]. We call attention to the confidence in the partner and how they guarantee they will not have any disease despite most of them being asymptomatic. This is a major problem that may justify a part of this gap between what they know and what they do, especially when they recognise false prevention measures like contraceptive pills or genital washing. Furthermore, it is possible that condoms are more associated with contraception than STI prevention, decreasing their utilisation when there are other concomitant contraceptive methods [31].

Likewise, better knowledge regarding STI prevention has been shown to decrease the importance given to sexual partners’ general appearance in the likelihood of sexual intercourse. Adolescents who place greater emphasis on superficial factors, such as appearance, could be more prone to engage in sexual intercourse without thoroughly evaluating the potential risks. Indeed, previous studies concluded that the more someone is attracted to a potential partner, the less they perceive that person as a health risk [32]. Once appearance is a primary consideration in sexual decision making, other critical and safety factors could be disregarded, namely, proper communication with sexual partners and a discussion about STI prevention. Besides that, individuals who fundamentally lack comprehensive knowledge about STI transmission and prevention may also be more susceptible to relying on these superficial factors in sexual decision making, creating a misleading correlation between appearance and sexual health status. In fact, previous studies show that respondents often report that they “just know” whether a sexual partner is safe or not only through their appearance [33], which certainly could lead to uninformed and risky sexual behaviours. Therefore, well-educated adolescents are more prone to focus their decision making on factors that have an actual impact on their sexual health [11]. In our study, better knowledge was associated with the importance given to the partner showing a negative HIV test, checking for lesions in the genital area, taking regular contraception and actual condom availability, meaning more attention being paid to specific indicators concerning STI prevention. Besides that, having these discussions about STIs with sexual partners reflects transparency and honest communication, which is an asset in sexual health. However, relying on negative test results could open the possibility of unprotected sex due to this sense of security, which could become a dangerous practice because of misleading results or negative HIV tests for a short period and even increasing the risk of contracting other STIs and unintended pregnancies. The aspect of asking for recent negative tests about HIV or other STIs deserves specific reflection. Screening for HIV, syphilis, gonorrhoea, and chlamydia infections is recommended by many institutions based on the benefit of the precocity of diagnosis in the attempted treatment and the possibility to cut the transmission chains, crucial for public health control of STIs [34,35]. However, the validity of the test is determined mostly by its positivity and not so much by its negativity, as it may well be in a latency period with the potential for transmission but with a (still) negative test. Therefore, the presence of a negative test does not provide sufficient information to confirm the non-existence of risk, which is why it must be contextualised in this sense and discouraged as a protector against STIs [36]. It is, therefore, a misconception and an opportunity for education.

The present study aims to address one of the weaknesses of previously developed studies in this area in terms of the sample, as it covers younger adolescents, specifically those in the ninth grade, just before the age where national studies reveal the onset of sexual activity. This allowed for a more in-depth analysis of the behaviour and knowledge of this group of young people, which is not possible in research studies covering a wide range of ages. This gives us the advantage of planning customised actions in middle school, particularly in promoting sexual education and intervention in primary healthcare settings.

One of the potential weaknesses of this study is the fact that it does not cover young people who have dropped out of school, which may constitute a selection bias in the study and possibly an overestimation of knowledge about sexually transmitted infections among adolescents. In the same way, the option to include students in their ninth-grade classes may bring some heterogeneity to the age of participants, with some of them being older than the 14 years expected for this class. Also, the choice of the municipality for participant selection is important, as it constitutes an urban area with a significant educational and cultural offering. Another question is related to the basic intellectual capacity of these adolescents. We know that greater academic resourcefulness can condition the way we learn and how we use available information, transforming it into attitudes and skills in all areas and also in health. However, the way to measure it is not consensual with most intelligence tests appearing more related to numeracy, verbal comprehension, memory, and information processing speed than with the transference of this knowledge into practice. That is why we went in search of knowledge in this area and how it is managed in attitudes towards concrete situations, regardless of academic success in mathematics or other subjects. Nevertheless, we think these constraints are not strong enough to call into question our conclusions.

## 5. Conclusions

In conclusion, our study reveals significant gaps in STI knowledge among 14-year-old adolescents in Portugal despite targeted prevention strategies such as condom use and HPV/HBV vaccination. Only a few preventive measures are widely recognised, and many misconceptions, like using antiseptics or oral contraception for STI prevention, persist. While HIV/AIDS, genital herpes, and hepatitis B are relatively well known due to media exposure, other STIs, like trichomoniasis and chlamydia, are less familiar. Factors influencing knowledge include sexual activity, emotional connection, and the influence of peers and the media. Reliable sources like healthcare providers, parents, and teachers are less frequently utilised than friends and the internet, which may lead to misinformation. Effective sexual education should focus on correcting misconceptions and promoting open communication about sexual health. Addressing these educational needs early, particularly in middle school, can help adolescents make informed decisions and reduce the risk of STI transmission.

## Figures and Tables

**Figure 1 healthcare-12-02283-f001:**
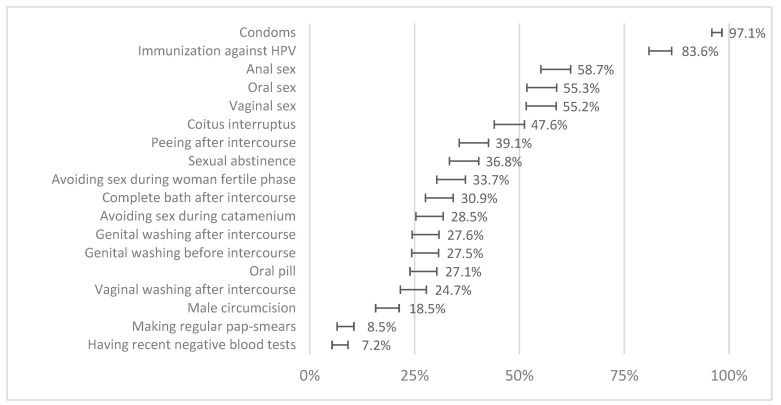
The proportion of correct answers to each protective measure against sexually transmitted infections (bars represent 95% confidence intervals).

**Figure 2 healthcare-12-02283-f002:**
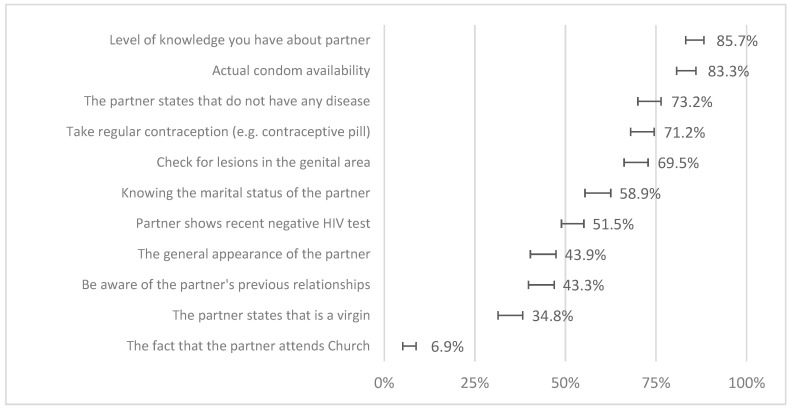
The importance of attitudes towards a partner’s attributes in the likelihood of having sexual intercourse (proportion of adolescents who these are important or very important in the total sample; bars represent 95% confidence intervals).

**Figure 3 healthcare-12-02283-f003:**
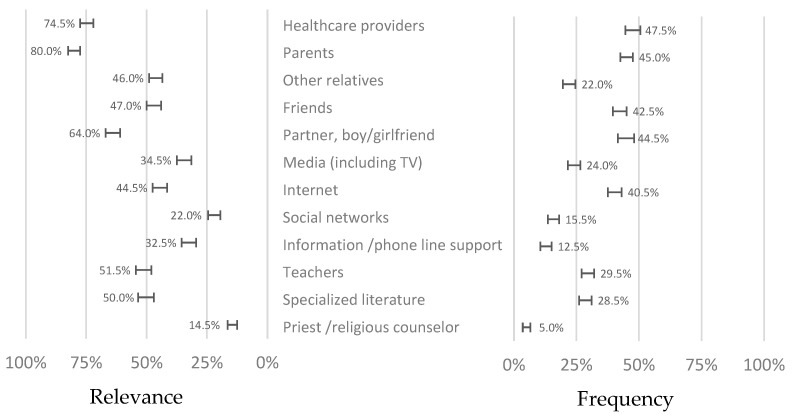
Relevance and frequency of utilisation of different sources of information about sexuality and sexually transmitted diseases for adolescents (bars indicate 95% confidence intervals).

**Table 1 healthcare-12-02283-t001:** General and demographic characteristics and sexual self-perceived knowledge of participants by gender.

Characteristics		Total Participants *n* (%)/Mean (±SD)	Sex *n* (%)/Mean (±SD)		
			Female	Male	*p* *
Number		750	382 (51.4)	361 (48.6)	
Age, mean ± SD		14.4 ± 1.0	14.3 ± 0.9	14.5 ± 1.1	0.005 *
Sexually active	Yes	121/640 (18.9)	53/341 (15.5)	67/297 (22.6)	0.024 *
Number of sexual partners in last year					
	0	6/103 (5.8)	3/44 (6.8)	3/58 (5.2)	0.335 *
	1	60/103 (58.3)	29/44 (65.9)	31/58 (53.4)	
	>1	37/103 (35.9)	12/44 (27.3)	24/58 (41.4)	
Self-perceived knowledge about sexuality issues					
	Mean	3.75 ± 0.72	3.68 ± 0.71	3.82 ± 0.73	0.030 **
	Insufficient	15/743 (2.0)	12/380 (3.2)	3/357 (0.8)	
	Reasonable	248/743 (33.4)	127/380 (33.4)	120/357 (33.6)	
	Good	382/743 (51.4)	208/380 (54.7)	171/357 (47.9)	
	Very good	98/743 (13.2)	33/380 (8.7)	63/357 (17.6)	
Words used to describe sexuality					
	Love	525/743 (70.7)	270/379 (71.2)	251/358 (70.1)	0.737 *
	Respect	511/743 (68.8)	275/379 (72.6)	233/358 (65.1)	0.028 *
	Pleasure	338/743 (45.5)	146/379 (38.5)	189/358 (52.8)	<0.001 *
	Health protection	203/743 (27.3)	124/379 (32.7)	78/358 (21.8)	<0.001 *
	Fidelity	184/743 (24.8)	95/379 (25.1)	89/358 (24.9)	0.949 *
	Freedom	159/743 (21.4)	79/379 (20.8)	79/358 (22.1)	0.686 *
	Prevention of pregnancy	117/743 (15.7)	62/379 (16.4)	55/358 (15.4)	0.712 *
	Self-esteem	104/743 (14.0)	65/379 (17.2)	38/358 (10.6)	0.011 *
	Open dialogue	75/743 (10.1)	44/379 (11.6)	30/358 (8.4)	0.145 *
	Information	54/743 (7.3)	26/379 (6.9)	28/358 (7.8)	0.617 *
	Beauty	33/743 (4.4)	9/379 (2.4)	24/358 (6.7)	0.005 *
	Impersonal sex	18/743 (2.4)	7/379 (1.8)	11/358 (3.1)	0.281 *
	Temperance	16/743 (2.2)	8/379 (2.1)	8/358 (2.2)	0.908 *
	Disengagement	8/743 (1.1)	3/379 (0.8)	5/358 (1.4)	0.428 *
Self-perceived knowledge about sexually transmitted infections					
	Mean	3.42 ± 0.80	3.42 ± 0.78	3.41 ± 0.81	0.544 **
	Insufficient	74/733 (10.1)	39/375 (10.4)	34/352 (9.7)	
	Reasonable	322/733 (43.9)	154/375 (41.1)	166/352 (47.2)	
	Good	284/733 (38.7)	162/375 (43.2)	121/352 (34.4)	
	Very good	53/733 (7.2)	20/375 (5.3)	31/352 (8.8)	
Self-perceived knowledge about specific diseases					
	HIV infection (AIDS)	4.09 ± 1.01	4.15 ± 0.94	4.06 ± 1.05	0.322 **
	Genital herpes	3.04 ± 1.32	3.13 ± 1.29	2.95 ± 1.36	0.064 **
	Hepatitis B virus	2.78 ± 1.31	2.90 ± 1.29	2.65 ± 1.34	0.013 **
	Hepatitis C virus	2.73 ± 1.33	2.84 ± 1.30	2.63 ± 1.35	0.026 **
	Human papillomavirus	2.27 ± 1.51	2.43 ± 1.56	2.12 ± 1.45	0.005 **
	Vaginal candidiasis	1.95 ± 1.58	2.22 ± 1.64	1.66 ± 1.46	<0.001 **
	Syphilis	1.69 ± 1.54	1.50 ± 1.54	1.86 ± 1.52	0.002 **
	Gonorrhoea	1.83 ± 1.46	1.70 ± 1.45	1.95 ± 1.47	0.027 **
	Chlamydia infection	1.09 ± 1.41	1.01 ± 1.39	1.15 ± 1.41	0.103 **
	Trichomonas infection	1.05 ± 1.30	1.01 ± 1.32	1.07 ± 1.25	0.237 **

* Chi-square test; ** Mann–Whitney U test.

**Table 2 healthcare-12-02283-t002:** Factors associated with better knowledge about sexually transmitted diseases.

		OR	95% CI	*p*
Male (vs. female) *		1.112	0.830–1.489	0.476
Age **		1.053	0.909–1.219	0.494
Sexually active ***		1.838	1.178–2.868	0.007
	≥1 sexual partner in the last year ***	0.766	0.114–5.143	0.784
Self-perceived knowledge about sexuality issues ***		1.378	1.116–1.701	0.003
Self-perceived knowledge about sexually transmitted infections ***		1.542	1.269–1.874	<0.001
Words used to describe sexuality ***				
	Love	1.408	1.018–1.950	0.039
	Respect	1.171	0.853–1.608	0.330
	Pleasure	1.332	0.991–1.790	0.058
	Health protection	1.093	0.787–1.518	0.595
	Fidelity	0.884	0.629–1.242	0.477
	Freedom	0.834	0.583–1.194	0.322
	Prevention of pregnancy	0.751	0.501–1.126	0.166
	Self-esteem	0.614	0.396–0.951	0.029
	Open dialogue	1.477	0.911–2.393	0.113
	Information	0.855	0.487–1.503	0.587
	Beauty	1.169	0.579–2.361	0.663
	Impersonal sex	0.981	0.382–2.520	0.967
	Temperance	0.462	0.145–1.470	0.191
	Disengagement	1.269	0.314–5.123	0.738
Self-perceived knowledge about specific diseases ***				
	HIV infection (AIDS)	1.304	1.113–1.526	<0.001
	Genital herpes	1.218	1.085–1.368	<0.001
	Hepatitis B virus	1.145	1.021–1.285	0.021
	Hepatitis C virus	1.082	0.966–1.211	0.172
	Human papillomavirus	1.131	1.023–1.253	0.019
	Vaginal candidiasis	0.993	0.898–1.098	0.708
	Syphilis	1.138	1.026–1.262	0.014
	Gonorrhoea	1.161	1.043–1.292	0.006
	Chlamydia infection	0.976	0.869–1.096	0.682
	Trichomonas infection	1.009	0.890–1.144	0.833
Frequency of appealing to sources of information adjusted to relevance ***				
	Healthcare providers	0.930	0.747–1.159	0.520
	Parents	0.881	0.695–0.118	0.297
	Other relatives	0.960	0.719–1.283	0.784
	Friends	1.419	1.090–1.847	0.009
	Partner, boy/girlfriend	1.025	0.801–1.311	0.847
	Media (including TV)	1.250	0.932–1.675	0.136
	Internet	1.293	0.981–1.704	0.068
	Social networks	0.816	0.566–1.175	0.274
	Information/phone line support	0.859	0.631–1.169	0.333
	Teachers	1.063	0.813–1.389	0.657
	Specialised literature	0.830	0.641–1.076	0.160
	Priest/religious counsellor	0.841	0.510–1.387	0.497

* Adjusted for age, ** adjusted for sex, and *** adjusted for age and sex.

**Table 3 healthcare-12-02283-t003:** The impact of knowledge on attitudes, measured by the importance given to the characteristics of the partner.

	OR *	95% CI	*p* **
The fact that the partner attends church	0.907	0.833–0.988	0.026
The partner states that they are a virgin	0.982	0.941–1.025	0.409
Be aware of the partner’s previous relationships	0.986	0.946–1.028	0.501
The general appearance of the partner	0.956	0.917–0.997	0.034
Partner shows recent negative HIV test	1.090	1.046–1.137	<0.001
Knowing the marital status of the partner	0.996	0.955–1.038	0.840
Checking for lesions in the genital area	1.060	1.013–1.101	0.011
Taking regular contraception (e.g., the pill)	1.099	1.048–1.153	<0.001
The partner states that they do not have any disease	0.984	0.940–1.031	0.500
Current condom availability	1.110	1.046–1.178	<0.001
Level of knowledge you have about partner	1.037	0.977–1.101	0.228

* Adjusted for age and sex; ** logistic regression model.

## Data Availability

The data are available by making direct contact with the authors, without undue reservation.

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
