# Peer review of "Adolescents’ Knowledge and Misconceptions About Sexually Transmitted Infections: A Cross-Sectional Study in Middle School Students in Portugal"

_healthcare, 2024, doi:10.3390/healthcare12222283_

Round 1

Reviewer 1 Report

Comments and Suggestions for Authors

Your study is interesting, though there are a few major shortcomings that really require correction. Firstly, check whether literature backs up your claims, because a few times I tried to follow through citations, I discovered serious mismatch between what the original paper claimed and what in your text you implied that was their conclusion. After this error repeated, I expect you also check the rest just in case, additionally you may try make a bit more nuanced interpretation. Secondly, there are a few issues where model part requires some extra attention.

LITERATURE:

Abstract:

“Yet, there are no studies evaluating their knowledge about sexually transmitted diseases and pre- vention methods at an earlier stage of their development.”

“Mendes, N., Palma, F., & Serrano, F. (2014). Sexual and reproductive health of Portuguese adolescents. International journal of adolescent medicine and health, 26(1), 3-12.”

Reis, M., Ramiro, L., Matos, M., Diniz, J., & Simões, C. (2011). Information and attitudes about HIV/AIDS in Portuguese adolescents: state of art and changes in a four year period.. Psicothema, 23 2, 260-6 .

I’d suggest more careful wording… (and check whether they are useful in your case as prior studies that partially included younger respondents)

Line 45 (though you mention this idea also in abstract).

“In Portugal, adolescents are initiating sexual activity at increasingly younger ages.”

“In recent years, the onset of sexual intercourse has occurred at increasingly younger ages (3)”

“3. Hansen BT, Kjaer SK, Arnheim-Dahlstrom L, Liaw KL, Juul KE, Thomsen LT, et al. Age at first intercourse, number of partners and sexually transmitted infection prevalence among Danish, Norwegian and Swedish women: estimates and trends from nationally 464

representative cross-sectional surveys of more than 100 000 women. Acta obstetricia et gynecologica Scandinavica. 2020;99(2):175-85. 465

DOI:10.1111/aogs.13732.

So you are reaching the conclusion that Portuguese teens become recently sexually active earlier… based on not that recent study (2005 vs 2012) covering… Scandinavia? 

Line 62:

“Early onset of sexual intercourse is related to lower knowledge (10) and to lower attitudes and skills, measured by less utilisation of condoms (7, 10), pri- marily due to method unavailability and false beliefs.

Therefore, literacy emerges as a fundamental factor in influencing decision-making 65

capacity in health, particularly in sexuality.”

Are you showing here causation or merely correlation?

I mean you can also demonstrate that early sexual onset is related to moderately low IQ in general, for example, though there is whole body of literature about it:

“Smart teens don’t have sex (or kiss much either)”

https://www.sciencedirect.com/science/article/abs/pii/S1054139X99000610

Please make this a bit more nuanced, as there are clearly more complicated factors, like fast life history...

Line 74:

However, it presents concerns about the quality of information and lack of scientific verification that limit its usefulness and reliability (13),

13. Brown-Johnson CG, Boeckman LM, White AH, Burbank AD, Paulson S, Beebe LA. Trust in Health Information Sources: Survey 487

Analysis of Variation by Sociodemographic and Tobacco Use Status in Oklahoma. JMIR Public Health Surveill. 2018;4(1):e8. 488

DOI:10.2196/publichealth.6260.

You are drawing conclusion on quality of online sex health advice for Portuguese teens based on survey which gauged what sources of health information random Oklahomans trust…

Line 359: 

Awaworyi Churchill S et al. found that Catholic doctrine may even deprive unmarried young people of adequate sex education (24).

24. Awaworyi Churchill S, Appau S, Ocloo JE. Religion and the Risks of Sexually Transmissible Infections: Evidence from Britain. 511

Journal of religion and health. 2021;60(3):1613-29. DOI:10.1007/s10943-021-01239-0.

Article in question:

“Our analyses suggest that being religious and frequently attending religious meetings are associated with lower odds of being diagnosed with STIs as well as lower self-assessed risk of getting HIV/AIDS. This seems particularly true for adherents of Christianity and more specifically, those of the Catholic denomination.”

While in such cases one can’t be certain whether they are all really uninfected or some are merely undiagnosed, I’d say that this study is interesting and should be included for more nuanced interpretation, except that it shows something completely different, that what you claim.

MODELING:

Line 221:

Although significative, the correlation with self-perception of knowledge was 222

very low (Pearson ρ = 0.139; p<0.001)

Pearson requires assumption that both variables follow normal distribution is quite unlikely for Likert scale questions. Use instead Kendall’s Tau.

Table 1:

Are you certain it adds up? You have 2 similar-sized populations of boys and girls, so average of whole sample should be somewhere between their average:

“C virus hepatitis 2.73 ± 1.33 2.71 ± 1.31 2.58 ± 1.40 0.026 **

Human papillomavirus 2.27 ± 1.51 2.24 ± 1.62 2.00 ± 1.48 0.005 **

Chlamydia infection 1.09 ± 1.41 0.98 ± 1.38 0.95 ± 1.40 0.103 **”

Average of whole sample is higher than for either boys or girls. Could you please check that? (and if a few of such values are quite improbable also check values that are within realm of possibility, just in case)

If there are some other valid reasons of this anomaly then state them in text.

Line 145:

“Methods deemed effective for STI transmission prevention included condom use, abstinence from sexual intercourse, male circumcision, and vaccination against Hepatitis B and HPV”

At least the way you presented it your delimitation appears somewhat arbitrary. 

condom / abstinence – OK

male circumcision, and vaccination against Hepatitis B and HPV – offers some protection,  one should not solely rely on it, though you marked it as CORRECT

“Having recent negative blood tests” – offers some protection (especially in ex. monogamous relationship), though you marked it as INCORRECT.

In order to run such questionnaire properly the question that you give to teens should be uncontroversial. Let’s see WHO:

“Appropriate sexually transmitted infections (STIs) diagnosis and treatment are crucial to prevent transmission, decrease deaths related to such infections (including still births and cervical cancer), while improving individual health, men’s and women’s sexual health, and the well-being of all people.”

Uhm… Crucial to prevent transmission...

https://www.who.int/teams/global-hiv-hepatitis-and-stis-programmes/stis/testing-diagnostics

For extra irony, those students who otherwise you deemed as more informed appear to even disagree with you on this point:

“Better knowledge is associated with higher importance given to actual condom availability, taking regular contraception, asking for a recent negative HIV test”

Either fortify your position with some empirical studies or guidelines that exactly show that in this edge case you made your delimitation correctly or change your answer grading to be logically consistent.

Line 327:

“We found that the most relevant sources of information about STIs were the healthcare providers, parents, and partners.”

Haven’t you found that the best predictor of actual knowledge were… friends (Apparently "asking for a friend" wasn't a made up excuse ;) )? I mean you found here something quite interesting and do not follow it through, while discussing in details sources that lacked predictive value in your model.

Author Response

Your study is interesting, though there are a few major shortcomings that really require correction. Firstly, check whether literature backs up your claims, because a few times I tried to follow through citations, I discovered serious mismatch between what the original paper claimed and what in your text you implied that was their conclusion. After this error repeated, I expect you also check the rest just in case, additionally you may try make a bit more nuanced interpretation. Secondly, there are a few issues where model part requires some extra attention.

Thank you for your comments, which allowed us to improve the paper and to rethink it in other directions previously not anticipated, such as the friends’ role in the peer sexual education.

LITERATURE:

Abstract:

“Yet, there are no studies evaluating their knowledge about sexually transmitted diseases and prevention methods at an earlier stage of their development.”

Mendes, N., Palma, F., & Serrano, F. (2014). Sexual and reproductive health of Portuguese adolescents. International journal of adolescent medicine and health, 26(1), 3-12.

Reis, M., Ramiro, L., Matos, M., Diniz, J., & Simões, C. (2011). Information and attitudes about HIV/AIDS in Portuguese adolescents: state of art and changes in a four year period.. Psicothema, 23 2, 260-6 .

I’d suggest more careful wording… (and check whether they are useful in your case as prior studies that partially included younger respondents)

The review of Mendes et al. cites the Fronteira I et al. 2009 study, including teenagers from secondary school and not basic school, the study of Matos M et al. (2008), including communities of African migrants living in poorer neighbourhoods in Portugal, and Frade A et al. 1991 report about sex education in Portugal. The study of Reis et al (2011) focused only on HIV transmission knowledge, and some of their conclusions most linked to our aim were already cited in reference 4.

They are not quite the same aim we had to study the knowledge about prevention of STD in general. Nevertheless, we changed the wording.

Line 45 (though you mention this idea also in abstract).

“In Portugal, adolescents are initiating sexual activity at increasingly younger ages.”

“In recent years, the onset of sexual intercourse has occurred at increasingly younger ages (3)”

“3. Hansen BT, Kjaer SK, Arnheim-Dahlstrom L, Liaw KL, Juul KE, Thomsen LT, et al. Age at first intercourse, number of partners and sexually transmitted infection prevalence among Danish, Norwegian and Swedish women: estimates and trends from nationally representative cross-sectional surveys of more than 100 000 women. Acta obstetricia et gynecologica Scandinavica. 2020;99(2):175-85. 465 DOI:10.1111/aogs.13732.

So you are reaching the conclusion that Portuguese teens become recently sexually active earlier… based on not that recent study (2005 vs 2012) covering… Scandinavia? 

We do not state, nor we could, that Portuguese adolescents had their sexual debut earlier. There are no studies to prove this point. The only study in the European region in the last years compares the four Scandinavian countries and reaches that conclusion. The Portuguese study of Reis et al. compares the years 2002 and 2006 but does not present the same result and does not allow that comparison, despite a lower proportion of sexual debut under 13 years. We changed the wording to clarify this comment.

Line 62:

“Early onset of sexual intercourse is related to lower knowledge (10) and to lower attitudes and skills, measured by less utilisation of condoms (7, 10), primarily due to method unavailability and false beliefs.

Therefore, literacy emerges as a fundamental factor in influencing decision-making capacity in health, particularly in sexuality.”

Are you showing here causation or merely correlation?

I mean you can also demonstrate that early sexual onset is related to moderately low IQ in general, for example, though there is whole body of literature about it:

“Smart teens don’t have sex (or kiss much either)”

https://www.sciencedirect.com/science/article/abs/pii/S1054139X99000610

Please make this a bit more nuanced, as there are clearly more complicated factors, like fast life history...

The question is not about intelligence or the measure of IQ, but the utilization each one makes from available information -literacy. We changed the text to make it clear.

Line 74:

However, it presents concerns about the quality of information and lack of scientific verification that limit its usefulness and reliability (13),

13. Brown-Johnson CG, Boeckman LM, White AH, Burbank AD, Paulson S, Beebe LA. Trust in Health Information Sources: Survey Analysis of Variation by Sociodemographic and Tobacco Use Status in Oklahoma. JMIR Public Health Surveill. 2018;4(1):e8. 488 DOI:10.2196/publichealth.6260.

You are drawing conclusion on quality of online sex health advice for Portuguese teens based on survey which gauged what sources of health information random Oklahomans trust…

We thank you for this comment, despite this is not a conclusion, just the rationale for the introduction. In a previous study, we found exactly the same conclusion with internet utilization conditioning negatively the literacy. (reference 14)

Line 359: 

Awaworyi Churchill S et al. found that Catholic doctrine may even deprive unmarried young people of adequate sex education (24).

24. Awaworyi Churchill S, Appau S, Ocloo JE. Religion and the Risks of Sexually Transmissible Infections: Evidence from Britain. Journal of religion and health. 2021;60(3):1613-29. DOI:10.1007/s10943-021-01239-0.

Article in question:

“Our analyses suggest that being religious and frequently attending religious meetings are associated with lower odds of being diagnosed with STIs as well as lower self-assessed risk of getting HIV/AIDS. This seems particularly true for adherents of Christianity and more specifically, those of the Catholic denomination.”

While in such cases one can’t be certain whether they are all really uninfected or some are merely undiagnosed, I’d say that this study is interesting and should be included for more nuanced interpretation, except that it shows something completely different, that what you claim.

Thank you for your comment. The citation was in the wrong sentence. It is corrected now.

MODELING:

Line 221:

Although significative, the correlation with self-perception of knowledge was very low (Pearson ρ = 0.139; p<0.001)

Pearson requires assumption that both variables follow normal distribution is quite unlikely for Likert scale questions. Use instead Kendall’s Tau.

Thank you for your attention. We corrected accordingly your comment.

Table 1:

Are you certain it adds up? You have 2 similar-sized populations of boys and girls, so average of whole sample should be somewhere between their average:

“C virus hepatitis 2.73 ± 1.33 2.71 ± 1.31 2.58 ± 1.40 0.026 **

Human papillomavirus 2.27 ± 1.51 2.24 ± 1.62 2.00 ± 1.48 0.005 **

Chlamydia infection 1.09 ± 1.41 0.98 ± 1.38 0.95 ± 1.40 0.103 **”

Average of whole sample is higher than for either boys or girls. Could you please check that? (and if a few of such values are quite improbable also check values that are within realm of possibility, just in case)

If there are some other valid reasons of this anomaly then state them in text.

Thank you for your attention. We corrected the errors in table 1, due to a wrong copy-paste.

Line 145:

“Methods deemed effective for STI transmission prevention included condom use, abstinence from sexual intercourse, male circumcision, and vaccination against Hepatitis B and HPV.

 At least the way you presented it your delimitation appears somewhat arbitrary. 

condom / abstinence – OK

male circumcision, and vaccination against Hepatitis B and HPV – offers some protection, one should not solely rely on it, though you marked it as CORRECT

“Having recent negative blood tests” – offers some protection (especially in ex. monogamous relationship), though you marked it as INCORRECT.

In order to run such questionnaire properly the question that you give to teens should be uncontroversial. Let’s see WHO:

“Appropriate sexually transmitted infections (STIs) diagnosis and treatment are crucial to prevent transmission, decrease deaths related to such infections (including still births and cervical cancer), while improving individual health, men’s and women’s sexual health, and the well-being of all people.”

Uhm… Crucial to prevent transmission...

https://www.who.int/teams/global-hiv-hepatitis-and-stis-programmes/stis/testing-diagnostics

For extra irony, those students who otherwise you deemed as more informed appear to even disagree with you on this point:

“Better knowledge is associated with higher importance given to actual condom availability, taking regular contraception, asking for a recent negative HIV test”

Either fortify your position with some empirical studies or guidelines that exactly show that in this edge case you made your delimitation correctly or change your answer grading to be logically consistent.

We completely agree with you that diagnosing STIs is essential for treating patients and cutting the chains of transmission, which naturally has a preventive impact.

Most STIs are asymptomatic at an early stage, and it is often recommended to carry out a screening test for HIV, chlamydia, and gonorrhoea, for example, for this purpose. The validity of the test is determined by its positivity and not so much by its negativity, as it may well be in a latency period with the potential for transmission but with a (still) negative test. Therefore, the presence of a negative test does not provide sufficient information to confirm the non-existence of risk, which is why it must be contextualized in this sense and discouraged as a protector against STDs. It is, therefore, a misconception and an opportunity for education.

We add this reflection to the discussion

Line 327:

“We found that the most relevant sources of information about STIs were the healthcare providers, parents, and partners.”

Haven’t you found that the best predictor of actual knowledge were… friends (Apparently "asking for a friend" wasn't a made up excuse ;) )? I mean you found here something quite interesting and do not follow it through, while discussing in details sources that lacked predictive value in your model.

We agree! We included a comment in the discussion.

Reviewer 2 Report

Comments and Suggestions for Authors

1. Lines 85-88 need to be distinctively made as research questions

2. In line 144, you mean "administration/staff" rather than "directions"? The wording used is confusing.

3. Line 176, replace adjustment variables with confounding variables.

4. Line 283-284, please rephrase as it reads as if the questions are the subject rather than the participants. 

5. The discussion is well-written and provides sufficient evidence to support the study's findings with findings from the literature. however, I believe that the first section describing the strong strategy of HBV prevention in Portugal belongs in the introduction rather than discussion. The introduction lacks an overview of STI prevention in Portugal and such information should be added.

Comments on the Quality of English Language

Editing of text is needed since grammatical errors are apparent throughout the manuscript.

Author Response

  1. Lines 85-88 need to be distinctively made as research questions

We changed the wording of objectives.

  1. In line 144, you mean "administration/staff" rather than "directions"? The wording used is confusing.

Yes, administration!

  1. Line 176, replace adjustment variables with confounding variables.

Done

  1. Line 283-284, please rephrase as it reads as if the questions are the subject rather than the participants. 

Done

  1. The discussion is well-written and provides sufficient evidence to support the study's findings with findings from the literature. however, I believe that the first section describing the strong strategy of HBV prevention in Portugal belongs in the introduction rather than discussion. The introduction lacks an overview of STI prevention in Portugal and such information should be added.

Done. We included the description in the introduction and changed the comments in the discussion.

Round 2

Reviewer 1 Report

Comments and Suggestions for Authors

Looks fine. 

As side note:

"They are not quite the same" I was not claiming that they are the same. I simply consider claim "no studies" as warning sign suggesting that either the article is going to be on some highly original and novel subject or that I should better check for other issues with literature review.

 "The question is not about intelligence or the measure of IQ, but the utilization each one makes from available information -literacy. We changed the text to make it clear."

You are building here the reasoning under implicit assumption - those teenagers lack knowledge, if only they had those information, they would behave like those more informed ones, right?

If you are testing for general knowledge, you are also capturing partially IQ and without further testing you don't really know whether those information were crucial or just had been correlates for general factor of intelligence. If you had been asking those teens for capital of South Korea, square root of 81 and year in which WW1 started, you should be also able to capture some statistically significant negative correlation with correct answers and risky sexual behaviors.

To control for that, you'd have to measure how much better predictor of behaviors had been knowledge on particular sexual subject vs for example general knowledge. Otherwise you'd be likely to overestimate impact of policy intervention.

Author Response

Looks fine.

Thank you very much for your attention and your comments. They allowed to review deeply our paper and to improve it.

As side note:

"They are not quite the same" I was not claiming that they are the same. I simply consider claim "no studies" as warning sign suggesting that either the article is going to be on some highly original and novel subject or that I should better check for other issues with literature review.

We agree and took your comment under attention when we changed the wording in the text

 "The question is not about intelligence or the measure of IQ, but the utilization each one makes from available information -literacy. We changed the text to make it clear."

You are building here the reasoning under implicit assumption - those teenagers lack knowledge, if only they had those information, they would behave like those more informed ones, right?

If you are testing for general knowledge, you are also capturing partially IQ and without further testing you don't really know whether those information were crucial or just had been correlates for general factor of intelligence. If you had been asking those teens for capital of South Korea, square root of 81 and year in which WW1 started, you should be also able to capture some statistically significant negative correlation with correct answers and risky sexual behaviors.

To control for that, you'd have to measure how much better predictor of behaviors had been knowledge on particular sexual subject vs for example general knowledge. Otherwise you'd be likely to overestimate impact of policy intervention.

We understand your rational for this discussion. The intelligence is far more complex than the IQ quotation, based mostly in the numeracy, verbal comprehension, memory and processing speed, with known issues about its applicability and real significance. Knowledge about health is not different from knowledge about any other issue and we agree that the facility to learn some contents is extendable to these topics. Also, although knowledge is relevant for development of the attitudes, is not mandatory as we figure from Fink’s theory of learning, which defends the thesis that there can exist specific skills without there being exact knowledge that supports them. That’s why we used questions both about knowledge and about attitudes to measure their relation. In fact, we did not prospect about intelligence (whatever the way to measure it) or academic achievements, but to be honest, we do not think it would change our conclusions.

Thank you for this comment. We included it on the limitations of our research.